# Nutritional and Morphofunctional Assessment in a Cohort of Adults Living with Cystic Fibrosis with or Without Pancreatic Exocrine and/or Endocrine Involvement

**DOI:** 10.3390/nu17132057

**Published:** 2025-06-20

**Authors:** Ana Piñar-Gutiérrez, José Luis Pereira-Cunill, Andrés Jiménez-Sánchez, Silvia García-Rey, María del Carmen Roque-Cuéllar, Antonio J. Martínez-Ortega, Irene González-Navarro, Esther Quintana-Gallego, Ángeles Pizarro, Francisco Javier Castell, Manuel Romero-Gómez, Pedro Pablo García-Luna

**Affiliations:** 1UGC Endocrinología y Nutrición, Hospital Universitario Virgen del Rocío, Avda. Manuel Siurot s/n, 41013 Seville, Spain; ana.pinar.sspa@juntadeandalucia.es (A.P.-G.);; 2UGC Endocrinología y Nutrición, Instituto de Biomedicina de Sevilla, IBiS/Hospital Universitario Virgen del Rocío/CSIC/Universidad de Sevilla, Avda. Manuel Siurot s/n, 41013 Seville, Spain; 3UGC Neumología y Cirugía Torácica, Instituto de Biomedicina de Sevilla, IBiS/Hospital Universitario Virgen del Rocío/CSIC/Universidad de Sevilla, Avda. Manuel Siurot s/n, 41013 Seville, Spain; 4UGC Aparato Digestivo, Instituto de Biomedicina de Sevilla, IBiS/Hospital Universitario Virgen del Rocío/CSIC/Universidad de Sevilla, Avda. Manuel Siurot s/n, 41013 Seville, Spain; 5UGC Radiodiagnóstico, Instituto de Biomedicina de Sevilla, IBiS/Hospital Universitario Virgen del Rocío/CSIC/Universidad de Sevilla, Avda. Manuel Siurot s/n, 41013 Seville, Spain

**Keywords:** cystic fibrosis, morphofunctional assessment, nutritional ultrasound, bioelectrical impedance, handgrip strength, GLIM criteria, cystic fibrosis-related diabetes, exocrine pancreatic insufficiency

## Abstract

**Objectives**: To describe the results of nutritional and morphofunctional assessment in a cohort of adults with cystic fibrosis; to evaluate differences in nutritional status between patients with and without exocrine and/or endocrine pancreatic involvement. **Methods**: Cross-sectional study: A cohort of adults with cystic fibrosis evaluated in a multidisciplinary unit was analyzed. Pancreatic status was examined, and malnutrition was diagnosed according to GLIM criteria. Morphofunctional assessment consisted of nutritional ultrasound, bioelectrical impedance, handgrip dynamometry, and anthropometry. Qualitative variables are expressed as *n* (%), quantitative variables as median (IQR). For group comparisons, Fisher’s exact test was used for qualitative variables and the non-parametric median comparison test for quantitative variables. **Results**: *n* = 101 participants were recruited, of whom 44 (43.6%) were women. Median age was 33 (25–40.5) years. A total of 64 participants (63.4%) had exocrine pancreatic insufficiency (EPI), 44 (43.6%) had endocrine pancreatic insufficiency, and 28 (27.7%) had cystic fibrosis-related diabetes (CFRD). Median BMI was 23.4 (20.1–24.89) kg/m^2^. A total of 48 patients (47.5%) were malnourished. Males with EPI had a higher prevalence of undernourishment than those without (56.4% vs. 16.7%, *p* = 0.005), but not women. CFRD patients displayed no differences in morphofunctional assessment. **Conclusions**: Almost half the sample was undernourished using GLIM criteria. Males with exocrine pancreatic insufficiency had worse nutritional status. Endocrine pancreatic involvement did not affect nutritional status.

## 1. Introduction

Cystic fibrosis (CF) is an autosomal recessive disease caused by a mutation in the gene on the long arm of chromosome 7 that encodes the CFTR (Cystic Fibrosis Transmembrane Conductance Regulator) protein involved in the passage of chloride ion across cell membranes. In people with normal CFTR function, chloride uptake occurs in the sweat glands. When CFTR dysfunction occurs, this uptake is disturbed and the concentration of chloride in sweat increases. In the airway, this leads to decreased chloride and bicarbonate secretion at the apical membrane and sodium uptake due to loss of CFTR-mediated sodium channel inhibition, leading to absorption of fluids from the airway surface. As a consequence, the composition of sweat and mucus is altered, thickening the latter and reducing mucociliary transport, leading to mucus retention and clogging of the airway, which becomes inflamed and more prone to infection [1]. Thicker fluids, therefore, lead to organ involvement, especially in the lungs, pancreas, and digestive tract [2].

Due to its low prevalence, it is considered a rare disease [3]. This pathology used to be lethal in childhood. Thanks to therapeutic advances in recent years, most patients are adults nowadays, even reaching the age of 50 in some cases [1]. Consequently, the clinical manifestations have changed, becoming more complex and leading to previously unknown long-term complications [4].

Malnutrition occurs in 25–30% of CF patients and is a marker of disease progression and a predictor of morbidity and mortality [5,6]. It is closely related to other clinical manifestations, especially pulmonary, and chronic infection and inflammation. Its etiology is based on three pillars: malabsorption losses (mainly due to exocrine pancreatic insufficiency: EPI), decreased intake (especially when there are complications associated with anorexia), and increased energy expenditure (especially due to increased work of breathing) [7].

The assessment of nutritional status has changed in recent years, with increasing importance being placed on muscle. A decrease in lean mass, which may be present despite a normal or high BMI, has been associated with a worse prognosis in CF [8,9,10]. Classically, it has been recommended to obtain weight, BMI, and intake assessment every 6 months in adults at every CF patient visit [10,11]. However, the current trend in Clinical Nutrition is to perform the so-called ‘morphofunctional assessment’ (MFA) of disease-related malnutrition in patients with different pathologies that affect nutritional status [12]. This consists of a combination of intake assessment, biochemical parameters, anthropometry, dynamometry, nutritional ultrasound, and bioelectrical impedance analysis (BIA).

The main objective of our study is to describe the results of nutritional and morphofunctional assessment in a cohort of adults with cystic fibrosis, and to evaluate the differences between patients with and without exocrine and/or endocrine pancreatic involvement.

## 2. Materials and Methods

### 2.1. Participants

This study is a sub-analysis of the one previously published in the Journal of Cystic Fibrosis [13]. The inclusion criteria were as follows:Age over 18 years.CF diagnosed by genetic test.Active follow-up at the Hospital Universitario Virgen del Rocío.Ability to understand and sign the study protocol and informed consent.The exclusion criteria were:Not having the capacity to understand and sign the study protocol and informed consent.Having an unstable clinical situation regarding their illness.Any circumstance that, in the opinion of the research team, interfered with the study protocol.

### 2.2. Study Design

A cross-sectional study was conducted in all patients in the adult CF cohort of the Hospital Universitario Virgen del Rocío who met the above criteria and signed the informed consent form of this study. The first participant was measured in January 2021 and the last in September 2022.

This study was approved by the Ethics Committee of the Hospital Universitario Virgen del Rocío.

### 2.3. Study Variables and Complementary Tests

All participants underwent this protocol. Regarding clinical assessment, age, presence of comorbidities, active treatments, and whether there was any consumption of toxic substances were collected in all participants.

#### 2.3.1. Pancreatic Function

The presence of exocrine pancreatic insufficiency (EPI) was assessed based on the use of lipase and/or the presence of significant steatorrhea and/or elastase levels, all assessed by a Digestive System specialist. The presence of endocrine pancreatic insufficiency was assessed following the criteria of the Cystic Fibrosis-Related Diabetes Clinical Care Guidelines [14], and classified into patients with CF-related diabetes mellitus (CFRD), CF-related prediabetes (CF-related prediabetes), and CF-related indeterminate glycaemia (CF-related indeterminate glycaemia). For that matter, fasting glucose, glycosylated hemoglobin (HbA1c), *C*-peptide, and the Homeostatic Model Assessment for Insulin Resistance (HOMA-IR) index were assessed in blood samples collected after an overnight fast.

#### 2.3.2. Other Laboratory Parameters

Blood count, visceral proteins (albumin, prealbumin, retinol-binding protein), renal function, ionogram (sodium, potassium, magnesium, phosphorus, plasma calcium), liver markers, iron metabolism, *C*-reactive protein, and a vitamin profile (vitamins A, E, D, B12, and folic acid) were also determined.

#### 2.3.3. Nutritional Status

Subjective global assessment (SGA) was performed to screen for malnutrition and for diagnosis, and a 6-month self-reported weight was registered to calculate weight loss percentage. Nutritional diagnosis was performed according to Global Leadership in Malnutrition (GLIM) criteria [15], while also incorporating the following MFA tests.

#### 2.3.4. Anthropometric Assessment

Weight in kilograms (kg) and height in meters (m) were determined using a digital scale Seca 769 with an incorporated stadiometer Seca 222 (seca GmbH & Co. KG, Hamburg, Germany). BMI was calculated as kg/m^2^. Waist circumference (WC) and mid-arm circumference (AC) were measured in centimeters (cm) with an anthropometric tape (Cescorf, Porto Alegre, Brazil). Tricipital skinfold (TS) was measured in millimeters (mm) using a Harpenden caliper (Baty International Ltd., Sheffield, United Kingdom). Mid-arm muscle circumference (MAMC) was calculated from AC and TS. P10 values between 30 and 39 years of age from a reference population were used [16].

#### 2.3.5. Bioelectrical Impedance Analysis (BIA)

A BIA, 101 BIVA (Akern Srl, Pontassieve, Italy), was used. Participants were placed in a supine position with arms separated at 30° and legs at 45°. The electrode placement area was cleaned with 70° alcohol. Low-impedance Ag/AgCl electrodes (Bivatrodes™ Akern Srl, Pontassieve, Italy) were placed proximal to the phalangeal-metacarpal joint on the dorsal surface of the right hand and distal to the transverse arch on the superior surface of the right foot. Resistance (R) in ohm (Ω), reactance (X_c_) in Ω, and phase angle (PA = arctan(X_c_/R) × 180°/*π*) in degrees (°) were registered. Fat-Free Mass Index (FFMI) was calculated from Fat-Free Mass built-in equations in Bodygram HBO version 3.1.6 (Akern SRL, Pontassieve, Florence, Italy) divided by the squared height of the participants.

#### 2.3.6. Handgrip Strength (HGS)

A JAMAR^®^ (Sammons Preston, Bolingbrook, IL, USA) dynamometer was used. Handgrip strength testing was performed with the participant seated and elbow flexed at 90° using his or her dominant hand, obtaining three measurements in kilograms (kg) with a one-minute rest in-between. Their mean value was used for statistical analysis, and dynapenia (low muscle strength) was defined using the P10 for the same sex and age categories of each participant [17].

#### 2.3.7. Nutritional Ultrasound [18]

A Diagnostic Ultrasound System, Model Z60 (Mindray, Hamburg, Germany), was used. Participants were placed in a supine position. Muscle ultrasound was performed at the level of the anterior rectus quadriceps, both statically and in relaxation. The measurement point was located at the lower third of an imaginary segment drawn between the anterosuperior iliac spine and the upper edge of the patella. Ultrasound for adipose tissue evaluation at the abdominal wall was performed at the midpoint between the xiphoid process and the umbilicus in the midline, with the patient also relaxed and in the supine position, taking a transverse image in unforced expiration. The following biomarkers were used for statistical analysis in this study:*Rectus femoris* muscle cross-sectional area (RF-CSA) in cm^2^. This parameter has demonstrated a significant correlation with other MFA parameters such as body cell mass (BCM) in BIA, HGS, and the Timed-Up and Go (TUG) test in patients with nutritional risk [19].Preperitoneal adipose tissue thickness (PPAT) in cm. Albeit with a slightly different methodology (measurement right below the xiphoid process), this parameter prospectively predicted insulin resistance in a cohort with high cardiovascular risk [20].

### 2.4. Statistical Analysis

Statistical analysis was performed using Statistical Package for Social Science (SPSS^®^) version 25 for Windows (IBM Corporation, Armonk, NY, USA). Quantitative variables are expressed using the median as a measure of centrality together with the interquartile range as a measure of dispersion, in median format (interquartile range). Qualitative variables are expressed as number of patients with data/total patients (percentage of total patients). For the comparison of groups, the Fisher test was used for qualitative variables and the non-parametric median comparison test for quantitative variables. Spearman’s coefficient (*r*) was used for correlation studies. The results of the univariate and multivariate analysis were expressed as odds ratio (OR) with 95% confidence interval (95% CI).

For the logistic regression, a univariate analysis was first performed on each variable that was considered as a potential malnutrition predictor, based on clinical judgment. Those variables that achieved a *p* < 0.1 were then included in Table 3. Then, these variables were evaluated again for a possible inclusion in the multivariate logistic regression model. Those with similar clinical significance were excluded: for example, age >45 years was not included as it was redundant having the time of evolution of cystic fibrosis; only fatty infiltration of body and tail of pancreas was included instead of the fatty infiltration of the head of the pancreas; and minimum function mutations were not included, since they were already considered under the exocrine pancreatic insufficiency (EPI) group. To avoid overfitting, the number of included variables for the multivariate analysis was further reduced: subjective global assessment, HbA1c level, and *C*-reactive protein levels were eliminated, as they had *p* > 0.05 in univariate analysis. Once this was accomplished, a stepwise logistic regression model was performed on the remaining variables. This whole process was performed with the support of a professional statistician.

A *p*-value of less than 0.05 was considered statistically significant.

## 3. Results

### 3.1. Demographics and Disease-Related Characteristics

A sample of 101 patients was obtained. Their demographics and main disease-related characteristics are shown in Table 1. Regarding chronic prescriptions, 64 patients (63.4%) took lipase and had EPI, 21 (20.8%) had insulin, 46 (45.5%) took CFTR modulators, and 42 (41.6%) consumed oral nutritional supplements.

### 3.2. Nutritional Status and Morphofunctional Assessment

A total of 48 participants (47.5%) were malnourished according to GLIM criteria. All met the etiological criterion of living with cystic fibrosis. Among these 48 patients, 39 cases (81%) met the phenotypic criteria for low muscle mass (diagnosed by low FFMI on BIA), 5 (10.4%) for BMI < 20 kg/m^2^, and 4 (8.3%) for weight loss of >5% in the last 6 months or >10% in more than 6 months. The prevalence of EPI was higher in malnourished participants (75% vs. 52.8%; *p* = 0.021), as seen in Appendix A. The results of the MFA performed are shown in Table 2. Up to 23 females (22.8%) reached the classic target BMI ≥ 22 kg/m^2^, and 42 males (41.6%) reached the classic target BMI ≥ 23 kg/m^2^. In total, 71 participants (70.3%) presented a normal BMI (18.5–25 kg/m^2^). Meanwhile, 6 cases (5.9%) had BMI < 18.5 kg/m^2^. A total of 21 patients (20.8%) were overweight, with 4 females (9.1%) and 17 males (29.8%). On the other hand, 3 patients (2.9%) were obese, with 1 female (2.3%) and 2 males (3.5%). Patients without EPI had a higher prevalence of being overweight, in contrast with those without EPI (35.1% vs. 12.5%; *p* = 0.007).

Among the 34 men with handgrip dynapenia, 18 (52.9%) also had reduced muscle mass (FFMI < 17 kg/m^2^). Of the 11 women with handgrip dynapenia, 4 (36.4%) also had reduced muscle mass (FFMI < 15 kg/m^2^).

### 3.3. Malnutrition-Related Clinical Factors

The factors associated with the diagnosis of malnutrition according to GLIM criteria are shown in Table 3.

### 3.4. Differences in Morphofunctional Assesment Between Patients with and Without Exocrine Pancreatic Involvement

Differences in MFA scores between patients with and without EPI involvement are shown stratified by sex in Table 4.

### 3.5. Differences in Morphofunctional Assesment Between Patients with and Without Endocrine Pancreatic Involvement

CFRD was present in 28 patients (27.7%). The time of diabetes evolution was 9.5 (4–15) years. Regarding their diabetes control, HbA1c was 6.4 (5.9–7.3)%. Their MFA results according to endocrine pancreatic involvement are shown in Table 5.

The results of the correlation between glycemic control (HbA1c) of patients with CFRD and quantitative results of MFA are shown in Table 6. The only nutritional biomarker that obtained a value close to significance was the FFMI value in men (*p* = 0.063), with a negative correlation (*r* = −0.552). There were also no differences in the HbA1c value between the groups of patients who met GLIM criteria and those who did not: 7 (5.8–7.6)% vs. 6.25 (5.92–7.27)%; *p* = 0.688.

HOMA-IR did not correlate with FFMI value in patients with CFRD (*r* = 0.028; *p* = 0.89) nor in patients without diabetes (*r* = 0.098; *p* = 0.428).

## 4. Discussion

To our knowledge, this is one of the studies with the largest number of adult CF patients undergoing MFA and in which the presence of malnutrition is evaluated according to GLIM criteria.

A total of 47.5% of patients met the GLIM criteria for malnutrition, contrasting with a high percentage of patients with normal BMI (18.5–25 kg/m^2^; 70.3%) and a relatively small prevalence of low BMI (<18.5 kg/m^2^; 5.9%). On the other hand, if the classic BMI target in CF patients (>22 kg/m^2^ in women and >23 kg/m^2^ in men) was considered, the percentages of patients reaching it were lower (22.8% and 41.6%, respectively). We consider that these discrepancies highlight the decreasing role of BMI as a nutritional biomarker in the face of MFA.

The latest ESPEN-ESPGHAN-ECFS guidelines on nutritional management of CF patients [10], published in 2024, recommend the use of BMI to assess whether a patient is underweight, overweight, or obese and to measure it at least every 3–6 months. However, for a complete assessment of nutritional status, a combination of clinical, anthropometric, biochemical, and body composition aspects is recommended. Specifically, it is recommended to assess fat mass (FM) and fat-free mass (FFM), because the association between these measurements and respiratory outcomes was stronger than that of BMI [21,22,23,24,25]. Another reason is that a low or normal BMI may mask alterations in FM or FMM. In our case, BMI was used to initially establish whether the patients met the phenotypic GLIM criteria. In those who did not meet this low BMI item, FFMI (FMM corrected for height) was evaluated. Regarding body composition analysis, a gold standard technique (DXA) was not available, so bioelectrical impedance analysis (BIA) was used instead. In the last nutritional management guidelines for CF [10], it is considered that BIA can be used in case of unavailability of DXA, although it has been previously described that BIA can overestimate FFMI compared to DXA, so the actual percentage of malnourished patients could be higher [26,27]. A limitation in its use is that it assesses body composition with predictive formulas that were not specific for this disease. On the other hand, electrolyte imbalances may affect this measurement, although our study was performed under conditions of clinical stability.

Regarding the prevalence of malnutrition in this study, it has been higher than in previous research in developed countries (4–19%) [10], yet this may be explained by different methods used to classify patients as malnourished. However, Sánchez-Torralvo et al. [28] obtained a similar prevalence of GLIM-based malnutrition (52.1%) in a cohort of 48 patients attended in Malaga (Andalusia, Spain) and with similar characteristics to ours, despite the fact that they used DXA to evaluate the FFMI (gold standard technique). Their percentage of women with low FFMI was higher than ours (61.1% vs. 38.6%), and lower in men (23.5% vs. 39.2%). In addition, the percentage of women in their study with malnutrition, according to GLIM criteria, was higher than ours (64% vs. 52.3%), while that of men was also lower (36% vs. 43.9%).

On the other extreme, there is growing concern about a possible increase in overweight and obesity in CF patients. This may be partly due to broad lifestyle changes across the population of developed countries. Before treatments with CFTR modulators could be used in a clinical setting; an increase of up to 345% in the prevalence of overweight and obesity had already been published in CF [29]. But specifically, the increase in adiposity in CF may be explained by improved prognosis and treatment options for this disease. In our study, obesity prevalence was low (2.9%), yet the percentage of overweight patients was remarkably high (20.8%)—especially in men (29.8%). In comparison, Petersen et al. [30] found obesity rates of 7.5–9.7% with the use of elexacaftor–tezacaftor–ivacaftor, and overweight rates of 19.4–31.3%. Before the era of CFTR modulators, the CF group in Malaga had a similar rate of normal BMI (67.8%), but a higher rate of underweight according to BMI (13.5% vs. 5.9%), and a lower rate of overweight (13.5% vs. 20.8%), in comparison with our results [31].

Our study has shown an association between exocrine pancreatic involvement and malnutrition, as previously described [10]. It is striking that these results achieved statistical significance in men, but not in women. Further studies in this regard should be carried out in the future to evaluate the reason for the difference between sexes, although it would be worth assessing whether male patients have less adherence to dietary and therapeutic measures than women, or whether this is due to sexual dimorphism. To date, we have found no evidence in this regard.

Regarding the differences in MFA between patients with different degrees of endocrine pancreatic insufficiency and those without it, in 2021 a study on 79 patients with CF and pancreatic insufficiency was published, with close resemblance to our cohort in terms of age, sex, and percentage of patients with CFRD [32]. Unlike our study, this previous research noted differences in BMI. On the other hand, these authors found a positive association between FFMI and both HOMA and fasting plasma insulin. However, we found no association between insulin resistance values and FFMI. These discrepancies may be explained by differences in FFMI assessment (DXA vs. BIA). In this regard, we found a nearly significant correlation between FFMI and HbA1c, although there were no differences in glycemic control between patients with and without malnutrition. In the future, it would be interesting to conduct more studies focused on evaluating whether muscle mass or function are related not only to the development of CFRD but also to its control or the need for higher insulin doses. In our study, patients with CFRD were generally well controlled, treated with insulin when necessary, and the HbA1c range of our cohort was narrow, which may have made it difficult to obtain significant results when comparing groups. In fact, other studies have already reported that patients with CFRD who initiate insulin treatment have a better nutritional status than those with indeterminate glycemia, which would therefore be consistent with our results [33].

Regarding the limitations of our study—in addition to its cross-sectional nature—it would have been interesting to add other functional tests to measure low physical performance. As future lines of research, we suggest to longitudinally evaluate the MFA in adults with CF to assess the prognostic value of different techniques (nutritional ultrasound, BIA, HGS), and evaluate how they may change with the use of CFTR modulators.

## 5. Conclusions

Patients with cystic fibrosis require specialized nutritional assessment and management. In our cohort of adults, despite adequate BMI levels, almost half of them presented malnutrition according to GLIM criteria due to low muscle mass. The factors associated with malnutrition were exocrine pancreatic insufficiency and longer disease evolution. Men with exocrine pancreatic insufficiency presented worse nutritional status, but there were no differences between patients with and without endocrine pancreatic insufficiency.

## Figures and Tables

**Table 1 nutrients-17-02057-t001:** Demographic and main characteristics related to their disease in a cohort of adult CF patients.

Variable	*n* (%)
Sex (women)	44 (43.6%)
Age (years) >45 years old	33 (25–40.5) 11 (10.9%)
Time of evolution (years)	24 (16–31)
Onset of symptomatology in infancy	75 (74.3%)
Type of mutation Minimal function Minimum/residual function Residual function	50 (49.5%) 48 (47.5%) 3 (3%)
ΔF508 mutation Homozygous Heterozygous	78 (77.2%) 29 (37.1%) 49 (62.8%)
FEV_1_	73 (52–90)
FVC	87 (74.5–97)
FEV_1_/FVC	69 (57–78)
Exocrine pancreatic insufficiency	64 (63.4%)
Endocrine pancreatic insufficiency CFRD CF-related prediabetes CF-related indeterminate glycemia	44 (43.6%) 28 (27.7%) 14 (13.9%) 2 (2%)
Malnutrition (GLIM criteria)	48 (47.5%)

*n*: absolute frequency; CF: cystic fibrosis; CFRD: CF-related diabetes; FEV_1_: forced expiratory volume in 1 s; FVC: forced vital capacity; GLIM: Global Leadership Initiative in Malnutrition.

**Table 2 nutrients-17-02057-t002:** Main results of morphofunctional assessment in a cohort of adult CF patients.

Variable	Whole Sample (*n* = 101)	Male (*n* = 57)	Female (*n* = 44)	*p* Between Sex
SGA (A)	93 (92%)	51 (89.5%)	42 (95.5%)	0.112
Adequate nutritional status (GLIM)	53 (52.5%)	32 (56.1%)	21 (47.7%)	0.401
Weight (kg)		71.6 (66.3–77.3)	55.8 (51.7–62.9)	<0.001
Height (cm)		173 (170–177.5)	161.5 (157–166.7)	<0.001
BMI (kg/m^2^)	23.4 (20.1–24.89)	23.96 (22.6–25.5)	22.1 (19.4–23.9)	0.01
Underweight ^a^	6 (5.9%)	4 (9.1%)	2 (3.5%)	0.239
Normal weight ^b^	71 (71.3%)	35 (79.5%)	36 (83.2%)	0.074
Overweight ^c^	21 (20.8%)	17 (29.8%)	4 (9%)	0.011
Obesity ^d^	3 (2.9%)	2 (3.5%)	1 (2.3%)	0.717
WC (cm)		86 (79.5–92.5)	74.5 (70–83.8)	0.003
Thresholded WC ^e^		52 (91.2%)	36 (81.8%)	0.152
AC (cm)		29 (27–30.8)	25 (23.5–28)	0.001
Thresholded AC ^f^		5 (8.8%)	4 (9.1%)	0.919
TS (mm)		11 (8.5–14)	13.4 (9–17)	0.286
Thresholded TS ^g^		12 (21.1%)	22 (50%)	<0.001
MAMC (cm)		25.4 (23.3–27.06)	20.68 (19.4–23.1)	<0.001
Thresholded MAMC ^h^		29 (28.7%)	25 (73.5%)	0.145
HGS (kg)		33.3 (27.4–40)	20 (17.7–23.3)	<0.001
Thresholded HGS ^i^		34 (59.6%)	11 (25%)	0.001
PA (°)		6.8 (6–7.1)	5.7 (5.3–6)	<0.001
FFMI (kg/m^2^)		17.8 (16.2–18.75)	15.55 (14.82–16.25)	<0.001
Thresholded FFMI ^j^		22 (39.2%)	17 (38.6%)	0.947
RF-CSA (cm^2^)		5.4 (4.2–6)	3.8 (2.8–4.2)	<0.001
PPAT (cm)		0.84 (0.62–1.15)	0.64 (0.48–0.94)	0.07

°: degrees; AC: mid-arm circumference; BMI: body mass index; FFMI: fat-free mass index; GLIM: Global Leadership In Malnutrition; HGS: handgrip strength; MAMC: mid-arm muscle circumference; P10: 10th percentile; PA: phase angle; PPAT: preperitoneal adipose tissue thickness; RF-CSA: *Rectus femoris* cross-sectional area, SGA: subjective global assessment; TS: tricipital skinfold; WC: waist circumference. ^a^ Underweight: BMI < 18.5 kg/m^2^; ^b^ Normal weight: BMI 18.5–25 kg/m^2^; ^c^ Overweight: BMI 25–30 kg/m^2^; ^d^ Obesity: BMI > 30 kg/m^2^; ^e^ Men < 102 cm, Women < 88 cm; ^f^ Men < P10 (25.86 cm), Women < P10 (22.6 cm); ^g^ Men < P10 (15.61 mm), Women < P10 (7.35 mm); ^h^ Men < P10 (22.29 cm), Women < P10 (15.92 cm); ^i^ Men < P10 (34.7 kg), Women < P10 (18 kg); ^j^ Men < 17 kg/m^2^, Women < 15 kg/m^2^.

**Table 3 nutrients-17-02057-t003:** Factors associated (*p* < 0.1) with the diagnosis of malnutrition according to GLIM criteria.

	Univariate Analysis		Multivariate Analysis	
Variable	OR (95% CI)	*p*	OR (95% CI)	*p*
BMI	0.719 (0.609–0.849)	<0.001		
EPI	2.679 (1.148–6.249)	0.023	3.148 (1.058–9.363)	0.039
FEV_1_	0.981 (0.963–0.999)	0.043		
Minimum function mutation	2.326 (1.047–5.168)	0.038		
HbA1c (%)	1.504 (0.935–2.418)	0.092		
SGA (A)	0.131 (0.015–1.135)	0.065		
Pancreatic fat infiltration, head (MRI)	1.011 (0.999–1.024)	0.078		
Pancreatic fat infiltration, body and tail (MRI)	1.015 (1.002–1.028)	0.024		
Age >45 years	0.213 (0.043–1.039)	0.056		
CF evolution time (years)	1.063 (1.017–1.111)	0.007	1.053 (1.003–1.105)	0.036
Elevated CRP	2.143 (0.893–5.144)	0.088		

95% CI: 95% confidence interval; BMI: body mass index; CF: cystic fibrosis; CRP: *C*-reactive protein; EPI: exocrine pancreatic insufficiency; FEV_1_: forced expiratory volume in 1 s; HbA1c: glycosilated hemoglobin; MRI: magnetic resonance imaging; OR: odds ratio; SGA: subjective global assessment.

**Table 4 nutrients-17-02057-t004:** Results of morphofunctional assessment in adults with cystic fibrosis according to the presence or absence of alterations in exocrine pancreatic function, stratified by sex.

	Men			Women		
Variable	Without EPI (*n* = 37)	With EPI (*n* = 64)	*p* Between EPI	Without EPI (*n* = 37)	With EPI (*n* = 64)	*p* Between EPI
GLIM +	3 (16.7%)	22 (56.4%)	0.005	9 (47.4%)	14 (56%)	0.570
SGA (A)	17 (94.4%)	34 (87.2%)	0.406	18 (94.7%)	24 (96%)	0.391
BMI (kg/m^2^)	25.1 (24.5–26.9)	23.4 (21.8–24.6)	0.001	22.3 (19.5–24.8)	19.2 (20.3–23.5)	0.543
Obesity	2 (11.1%)	0 (0%)	0.034	0 (0%)	1 (4%)	0.378
Overweight	9 (50%)	8 (20.5%)	0.024	4 (21.1%)	0 (0%)	0.016
WC (cm)	88.7 (85–96.1)	82.5 (78–90.4)	0.045	77 (72.2–87.3)	74 (68.8–81.8)	0.350
AC (cm)	30.7 (29.3–32.7)	27.7 (27–29.2)	0.394	27.5 (25–30)	25 (22.2–27.3)	0.266
TS (mm)	12.5 (7.6–15.1)	11 (8.5–13.2)	0.622	14 (8–16.2)	13.4 (9–17.2)	1
HGS (kg)	36.1 (32.9–45.3)	32 (32.7–36)	0.037	20 (16.6–26.6)	19.7 (17.9–23.3)	0.934
PA (°)	6.8 (6.1–7.2)	6.8 (5.8–7.1)	0.821	5.8 (5.5–6)	5.6 (5.3–5.9)	0.683
FFMI (kg/m^2^)	18.7 (18.1–20.7)	16.8 (15.5–17.9)	0.001	15.8 (15–16.7)	15.4 (14.5–15.6)	0.068
RF-CSA (cm^2^)	5.8 (4.8–6.9)	5.1 (3.9–5.7)	0.386	3.8 (2.9–4.1)	3.6 (2.7–4.7)	0.940
PPAT (cm)	0.9 (0.7–1.4)	0.7 (0.6–1.1)	0.339	0.58 (0.43–0.89)	0.64 (0.48–1.05)	0.959

°: degrees; AC: mid-arm circumference; BMI: body mass index; EPI: exocrine pancreatic insufficiency; FFMI: free fat mass index; HGS: handgrip strength; PA: phase angle; PPAT: preperitoneal adipose tissue thickness; RF-CSA: *Rectus femoris* cross-sectional area, SGA: subjective global assessment; TS: tricipital skinfold; WC: waist circumference.

**Table 5 nutrients-17-02057-t005:** Results of morphofunctional assessment in adults with cystic fibrosis according to the presence or absence of alterations in endocrine pancreatic function, stratified by sex.

Men					
Variable	Without Endocrine Pancreatic Insufficiency (*n* = 35)	CF-Related Indeterminate Glycaemia (*n* = 1)	CF-Related Prediabetes (*n* = 8)	CFRD (*n* = 13)	*p* Between Diagnosis
GLIM +	13 (37.1%)	0 (0%)	4 (50%)	8 (61.5%)	0.263
SGA (A)	32 (91.4%)	1 (100%)	7 (87.5%)	11 (84.6%)	0.867
BMI (kg/m^2^)	23.8 (22–25.8)	21 (16.9–21)	20.5 (19.6–23.3)	23.2 (19.7–24.5)	0.134
Obesity	2 (5.7%)	0 (0%)	0 (0%)	0 (0%)	1
Overweight	12 (34.3%)	1 (100%)	1 (12.5%)	3 (23.1%)	0.301
WC (cm)	86 (79–92)	86.5 (86.5–86.5)	83 (75.2–96)	87.5 (82.2–94.2)	0.348
AC (cm)	29.2 (27.5–31.8)	28 (28–28)	27 (25–30.5)	28.5 (26–29.7)	0.394
TS (mm)	11.5 (7.8–15)	9.8 (9.8–9.8)	11 (8.5–20)	11 (8.5–13)	0.461
HGS (kg)	35 (27.6–43.3)	35.3 (35.5–35.3)	31.5 (25.7–35.1)	31.7 (24.3–34.3)	0.273
PA (°)	6.9 (6.4–7.2)	5.9 (5.9–5.9)	6.6 (6.1–7)	6.6 (5.5–6.9)	0.46
FFMI (kg/m^2^)	17.8 (16.5–19.1)	19.2 (19.2–19.2)	17.8 (15–18.6)	16.5 (15.5–17.95)	0.555
RF-CSA (cm^2^)	5.3 (4.6–6.1)	4.8 (4.8–4.8)	4.4 (3.7–6.1)	5.6 (4.2–5.8)	0.521
PPAT (cm)	0.8 (0.6–1.2)	0.5 (0.5–0.5)	0.6 (0.5–1.1)	0.8 (0.6–1.1)	0.460
**Women**					
**Variable**	**Without Endocrine Pancreatic Insufficiency** **(*n* = 22)**	**CF-Related Indeterminate glycaemia** **(*n* = 1)**	**CF-Related Prediabetes** **(*n* = 6)**	**CFRD** **(*n* = 15)**	***p* Between Diagnosis**
GLIM +	10 (45.5%)	1 (100%)	5 (83.3%)	7 (46.7%)	0.325
SGA (A)	22 (100%)	1 (100%)	6 (100%)	14 (93.3%)	0.512
BMI (kg/m^2^)	22.1 (19.5–24.4)	17 (17–17)	19.8 (19.2–24.1)	22.6 (19.4–23.6)	0.531
Obesity	1 (4.5%)	0 (0%)	0 (0%)	0 (0%)	1
Overweight	3 (13.6%)	0 (0%)	1 (16.7%)	0 (0%)	0.322
WC (cm)	77 (71.5–86.2)	64 (64–64)	73 (67.2–74.7)	74.5 (69.6–83.7)	0.288
AC (cm)	25 (24.2–29.2)	21 (21–21)	25 (25–29.2)	25 (22.3–28)	0.796
TS (cm)	15.2 (9–17.7)	9 (9–9)	13.5 (6.5–14)	12.1 (9–16)	0.405
HGS (kg)	19.8 (16.6–23.9)	18.6 (18.6–18.6)	21 (20.1–23.7)	19.7 (17.3–23.6)	0.251
PA (°)	5.8 (5.3–6.2)	5.5 (5.5–5.5)	5.7 (5.4–6.1)	5.6 (5.3–5.9)	0.559
FFMI (kg/m^2^)	15.7 (14.9–16.9)	14.2 (14.2–14.2)	15.1 (14.4–16.3)	15.5 (14.6–15.6)	0.531
RF-CSA (cm^2^)	3.8 (3.1–4.1)	3.7 (3.7–3.7)	5 (3–6.5)	3.3 (2.7–3.9)	0.619
PPAT (cm)	0.6 (0.4–0.9)	0.5 (0.5–0.5)	0.5 (0.3–0.7)	0.7 (0.56–1.05)	0.399

°: degrees; AC: mid-arm circumference; CF: cystic fibrosis; CFRD: CF-related diabetes; FFMI: free fat mass index; HGS: handgrip strength; PA: phase angle; PPAT: preperitoneal adipose tissue thickness; RF-CSA: *Rectus femoris* cross-sectional area; SGA: subjective global assessment; TS: tricipital skinfold; WC: waist circumference.

**Table 6 nutrients-17-02057-t006:** Correlation between glycemic control (defined as glycosylated hemoglobin levels) and morphofunctional assessment results in patients with cystic fibrosis-related diabetes.

	HbA1c (%) in Men with CFRD (*n* = 13)		HbA1c (%) in Women with CFRD (*n* = 15)	
	*r*	*p*	*r*	*p*
BMI (kg/m^2^)	0.109	0.736	−0.170	0.580
WC (cm)	−0.155	0.631	0.130	0.672
AC (cm)	−0.400	0.223	0.012	0.970
TS (mm)	−0.303	0.338	−0.006	0.987
HGS (kg)	−0.489	0.107	−0.190	0.534
PA (°)	−0.362	0.248	−0.293	0.355
FFMI (kg/m^2^)	−0.552	0.063	−0.141	0.646
RF-CSA (cm^2^)	0.390	0.210	−0.281	0.352
PPAT (cm)	−0.257	0.420	0.254	0.427

°: degrees; AC: mid-arm circumference; CFRD: cystic fibrosis-related diabetes; FFMI: free fat mass index; HbA1c: glycosylated hemoglobin; HGS: handgrip strength; PA: phase angle; PPAT: preperitoneal adipose tissue thickness; RF-CSA: *Rectus femoris* cross-sectional area; SGA: subjective global assessment; TS: tricipital skinfold; WC: waist circumference. *r*: Spearman’s correlation coefficient.

## Data Availability

The data presented in this study are available on request from the corresponding author as per European legislation on data protection.

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
