# Peer review of "Nutritional and Morphofunctional Assessment in a Cohort of Adults Living with Cystic Fibrosis with or Without Pancreatic Exocrine and/or Endocrine Involvement"

_nutrients, 2025, doi:10.3390/nu17132057_

Round 1

Reviewer 1 Report

Comments and Suggestions for Authors

The authors conducted a cross-sectional study aimed at describing the results of nutritional and morphofunctional assessments in a cohort of 101 adults with cystic fibrosis.

Here are my suggestions:

1.

In Table 3, the authors present factors associated (p < 0.1) with the diagnosis of malnutrition according to the GLIM criteria, with effect sizes reported as odds ratios. Malnutrition was identified in 48 out of 101 participants (47.5%), as shown in Table 1. However, demographic or baseline characteristics stratified by malnutrition status (malnutrition vs. non-malnutrition) were not provided. The table includes only univariate and multivariate odds ratios, making it difficult for readers to interpret the distribution and differences between the two groups (malnutrition: n = 48; non-malnutrition: n = 53). Moreover, it remains unclear whether the multivariate analysis—expressed as odds ratios (ORs) with 95% confidence intervals (CIs)—was based on a stepwise logistic regression method, or whether only statistically significant variables (e.g., EPI and time evolution of cystic fibrosis) were retained in the model, with non-significant variables excluded without explanation.

2.

When conducting a multiple logistic regression analysis, a commonly recommended rule of thumb is to have at least 10 events per predictor variable (EPV) to avoid overfitting and unstable estimates. In Table 3, given that the study included 101 participants and 11 independent variables in the multiple logistic regression model, there is a risk of overfitting and unstable estimates due to a low events-per-variable ratio (approximately 4.4, calculated as 48 events divided by 11 variables). The authors are advised to justify the variable selection process, consider using dimension reduction or regularization techniques, and report diagnostic tests for model fit (e.g., Hosmer-Lemeshow test, variance inflation factors).

3.

Moreover, it remains unclear whether the multivariate analysis expressed as odds ratio (OR) with 95% confidence interval (95% CI) was conducted using a multiple logistic regression model. If so, did the authors apply a stepwise multiple logistic regression method? Or did they retain only statistically significant variables—such as EPI and time evolution of cystic fibrosis—in the model, while excluding non-significant variables without providing any explanation?

4.

Tables 4 and 5 present the results of the morphofunctional assessment in adult men and women with cystic fibrosis, according to the presence or absence of alterations in exocrine pancreatic function. Given the similarity in content and structure, it may be more effective to consolidate these into a single Table to facilitate comparison and improve clarity for readers.

5.

Tables 4 and 5 include two groups: without EPI and with EPI. It is recommended that the sample size for each group be provided to enhance clarity and allow readers to better interpret the results.

6.

Tables 4 and 5 present several variables—such as waist circumference (WC) and arm perimeter (AP)—that were not included in Tables 1 or 3. It is unclear whether these variables are specific to the assessment of exocrine pancreatic insufficiency (EPI), and clarification from the authors would be helpful.

7.

Tables 6 and 7 include four groups. It is recommended that the sample size for each group be provided to enhance clarity and help readers better interpret the results.

8.

Tables 6 and 7 include four groups. If some groups—for example, the CF-related indeterminate glycaemia group—have very small sample sizes, it may be advisable to combine certain groups. Small sample sizes can undermine the validity of statistical tests, potentially rendering the results less meaningful.

9.

Tables 6 and 7 present the results of the morphofunctional assessment in adult men and women with cystic fibrosis, stratified by the presence or absence of alterations in pancreatic endocrine function. Given the similarity in content and structure, consolidating these into a single table may enhance clarity and facilitate easier comparison for readers.

10.

In Table 8, the authors present the correlation between glycemic control and morphofunctional assessment results in patients with cystic fibrosis-related diabetes. However, it is unclear whether these correlations are presented according to sex. It is also not clear which variables are considered as measures of glycemic control, as the authors do not clearly specify this.

Moreover, since sex is included in Table 8, it is recommended that the authors clearly report the number of male and female patients with cystic fibrosis-related diabetes. The authors should specify the sample sizes for each sex group. If the sample sizes are too small, it would be more appropriate to use a nonparametric test, such as the Spearman correlation, rather than the Pearson correlation, to avoid potential violations of normality assumptions.

Additionally, it remains unclear how the correlation coefficients were calculated. For example, in men, a correlation coefficient (r) of 0.109 is reported, but it is not specified which variables are being correlated—for instance, is this the correlation between BMI and glycemic control, or between BMI and another variable? The authors are advised to clarify the variables involved in each reported correlation to improve interpretability.

Author Response

Dear colleague, 

Thank you for your comments. Please find a detailed response in the attached PDF file. 

Kind regards,

Reviewer 2 Report

Comments and Suggestions for Authors

This is a well-structured and clearly written cross-sectional study investigating the nutritional and morphofunctional status in a large adult cystic fibrosis (CF) cohort. The authors used validated tools (GLIM criteria, MFA techniques) and provide meaningful comparisons between subgroups based on pancreatic involvement. The findings are clinically relevant, especially in the context of growing adult CF populations and evolving treatment landscapes with CFTR modulators.

The paper offers important insights regarding the limitations of BMI as a sole nutritional marker and emphasizes the role of more comprehensive tools like bioelectrical impedance, handgrip strength, and nutritional ultrasound in detecting malnutrition in CF patients.

  1. Clarification of inclusion timeline: The manuscript states, “The first participant was measured on X and the last on Y.” Please provide exact dates.
  2. Definition of nutritional ultrasound markers: While the methods are clear, a brief justification for selecting RF-CSA and PPAT as key indicators would strengthen the rationale.
  3. Discussion on sex differences: The significant difference in malnutrition between males with EPI versus females warrants a deeper exploration or hypothesis (beyond adherence or dimorphism).
  4. Endocrine dysfunction impact: While the paper concludes no difference with CFRD, a more detailed subgroup analysis could improve clarity (e.g., comparing CFRD vs. prediabetes vs. normoglycemic CF patients).
  5. Use of GLIM criteria: It would be valuable to explain whether phenotypic and etiologic GLIM components were both met and how FFM/BIA was prioritized.

Minor Points

  • Line 26: "nutritional ultrasound®" – Consider using generic terminology unless branding is required.
  • Line 279–280: BMI thresholds should be explicitly stated in results or methods to clarify definitions (e.g., >22 kg/m² in women).
  • Table 1: Include BMI range (min–max) and BMI classification (normal, overweight, etc.) for greater context.
  • Abbreviations: Some abbreviations (e.g., PPAT, RF-CSA) should be defined upon first mention in the abstract or main text.
  • Language: Minor grammatical revisions throughout (e.g., "a result that was not obtained in ours" → “which was not observed in our cohort”).

This is a high-quality manuscript with important clinical implications. With minor clarifications and refinements, it will be suitable for publication.

Author Response

(The authors gave the same response as above.)

Round 2

Reviewer 1 Report

Comments and Suggestions for Authors

All the concerns have been answered.